# Familiarity Determines Whether Accent Affects Attitudes and Behaviors of the Listener

**DOI:** 10.3390/bs14060430

**Published:** 2024-05-22

**Authors:** Zenghu Cheng, Yugui She, Junjun Fu, Wenming Xu

**Affiliations:** 1School of Psychology, South China Normal University, Guangzhou 510631, China; 2019010211@m.scnu.edu.cn; 2Psychological Counseling Center, Jiangxi Vocational College of Mechanical & Electrical Technology, Nanchang 330045, China; sheyugui@outlook.com; 3School of Psychology, Key Laboratory of Behavioral and Mental Health of Gansu Province, Northwest Normal University, Lanzhou 730070, China; psyfjj@nwnu.edu.cn; 4Psychological Counseling Center, Jiaying University, Meizhou 514015, China

**Keywords:** accent, familiarity, stranger, acquaintance

## Abstract

Previous research found that accents cause the listener to exhibit prejudice toward the speaker. The present study tested whether the familiarity of the listener and speaker moderated this effect. Study 1 tested this question in a simulated recruit scenario and found that participants were less likely to recruit candidates with an accent, but this effect existed only when the candidate was a stranger to the interviewer, not when the candidate was an acquaintance. Study 2 retested this question in a scenario of talking one-on-one and also found that the effect of accent existed only when they were strangers, not when they were acquaintances. Both studies suggested that the effect of accent on the attitude and behavior of the listener vanished when the speaker and listener were familiar with each other. This work offers insights for understanding the effect of accent on social interaction from the perspective of the familiarity of the speaker and listener and reveals the moderated role of familiarity in the dynamic of the effect of accent.

## 1. Introduction

The accent, a distinctive way in which a group of people pronounce words [1], is not only a common linguistic phenomenon but also an important factor affecting our social interactions. The accents of speakers have a significant impact on how they are perceived by others, including their social status, sense of unity, and energy [2]. Research conducted by Aidil-Karanain revealed that individuals generally prefer speakers with native accents over those with foreign accents [3]. Additionally, Hooft discovered that people tend to find accents that resemble their own more attractive, as demonstrated through a study where participants rated the attractiveness of various English accents such as general American, standard British, Australian, and New Zealand [4]. Apart from influencing attitudes, accents can also influence how others behave towards the speaker. For example, a study by Deprez-Sims and Morris demonstrated that American individuals were more likely to hire individuals with American accents than those with French accents [5]. Similarly, Heblich et al. found that participants were more inclined to cooperate with individuals who spoke their home region’s accent compared to those who spoke accents from other regions [6].

However, the effect of accent on listeners’ attitudes or behaviors was not presented in all situations. There have also been studies that indicate no impact of accents on listeners’ attitudes or behaviors. Souza and Markman conducted research and found no significant differences between foreign-accented and native-accented speakers in terms of how intelligible, credible, and favorable they were perceived by native listeners [7]. Another study by Nejjari et al. involved German, Spanish, and Singaporean participants who listened to speeches in Dutch-accented English, standard British English, and standard American English separately [8]. Surprisingly, the researchers found that Dutch-accented English was not only as well understood as standard English, but it was actually evaluated just as positively by German listeners and even more positively than standard English by Spanish and Singaporean listeners.

What enhances the influence of accent, and what reduces the influence of accent on social interaction? Kang et al. found that the situational contexts in which speakers are heard play a pivotal role in influencing how listeners perceive their accented speech. When speakers are heard in service-oriented occupational settings, they are judged to be much more intelligible and acceptable compared to when they are heard in academic environments [9]. Bresnahan et al. argue that ethnic identity could influence the effect of accent. American individuals who display a robust sense of ethnic identity tend to favor American English, whereas those with a less pronounced ethnic identity are more open to accepting foreign accents [10].

Besides these, one of the potential factors is whether the speaker is a stranger or an acquaintance of the listener. Previous studies have suggested that familiarity has a significant effect on social interaction. Freeberg, for instance, has indicated that familiarity is fundamental to communication and interaction in social species [11]. Reis et al. also found that familiarity led to attraction in live interactions between previously unacquainted individuals [12]. Yamane explores how psychological distance affects interpersonal experience, finding that people may find it difficult to treat peripheral acquaintances as true acquaintances [13]. Schroeder et al. argued that individuals tended to engage in fewer enjoyable interactions with strangers due to their own avoidance of such conversations, thereby creating a self-fulfilling prophecy regarding their social experiences [14]. These studies suggest that familiarity promotes positive social interactions and may be a bridge to social relationships. But Savitsky et al. argued that people might not really communicate better with friends than with strangers; they just overestimate how well they communicate with friends [15]. Beery and Shambaugh also found that preference for familiarity may be linked to social groups that are more insular and exhibit less flexibility in their social dynamics [16]. Overall, familiarity plays an important role in social interaction. The present study aimed to explore whether familiarity moderates the effect of accent on social interaction.

A regional accent is a type of accent that is specific to a particular geographical area. China has numerous regional accents, one of which is the accent which is commonly used by the native population of Henan province. Individuals with the Henan accent pronounce words in a distinct manner compared to the mainstream Mandarin accent, the standard accent in China. However, those with a Henan accent essentially use the same vocabulary, grammar, and syntax as the mainstream Mandarin accent. Thus, the Henan accent can be considered an “accent”, not a “dialect”. The biggest difference between the Henan accent and the mainstream Mandarin accent is tones. The mainstream Mandarin accent has four standard tones, while the Henan accent employs some additional tonal variations or differences in tones. For example, where the third tone should be used in a mainstream Mandarin accent, the first tone is often used in a Henan accent. However, despite the difference in pronunciation, individuals from other regions can still comprehend the Henan accent, and research has reported no obvious stigma related to the Henan accent, so we selected the Henan accent as a regional accent and the mainstream Mandarin accent as a standard accent in the present study.

For this research, we conducted two studies aiming to offer a distinct perspective on understanding whether familiarity influences the impact of accents on listeners’ attitudes toward the speaker. In Study 1, we first explored whether the effect of accent on hiring decisions was modulated by familiarity. In Study 2, we again tested these effects in the situation of talking with an acquaintance or a stranger one-on-one.

## 2. Study 1

### 2.1. Method

#### 2.1.1. Participants

Twenty-six Chinese college students (24 males; all participants spoke with a mainstream Mandarin accent) took part in the experiment. In order to determine the sample size required for the present studies, an a priori effect analysis (ANOVN: repeated measure, α = 0.05, effect size = 0.25, 1 − β = 0.8) was conducted using the G*Power 3.1 program, and the suggested least sample size was 24. Thus, according to this suggestion, the available resources, and the possible exclusion, the sample sizes of the two studies were 26. The participants came from one college class. There were also four associates of the experimenter. Two of them were from the same college class as the participants, and the others were from another college class and were strangers to all the participants. This study was approved by the research ethics committee at our university and conducted in accordance with the principles of the Declaration of Helsinki. All participants in this study were paid for the experiment. Written informed consent was obtained from all the participants.

Since the two acquainted associates were familiar with each participant, to exclude the influence of their relationships, we asked the associates to rate their relationship with each participant and counterbalanced this factor across conditions (mainstream Mandarin accent and Henan accent).

#### 2.1.2. Procedure and Materials

The participants were told that the experiment aimed to study the interview process, and that it would involve a simulated online structured interview for the college’s painting society. They were also told that they would draw lots to decide whether they would be interviewers or candidates, and that those designated as candidates would draw further lots to decide whether they would speak with a mainstream Mandarin accent or with a regional (most Chinese can speak both with a mainstream Mandarin accent and with a hometown accent). In reality, this was not true: the setup was designed to reduce any odd feelings among participants upon hearing their classmates speak in a Henan accent. In fact, the experiment was arranged in such a way that all the regular participants were designated as interviewers; only the associates of the experimenter were designated as candidates, and in the actual experiment they would speak with either a mainstream Mandarin or a Henan accent.

The interviews were one-on-one and conducted online. Via a screen, the participants saw the candidate come into a room and sit in front of a desk (this was actually a pre-recorded video). The experimenter then told the candidate to pick up the piece of paper on the desk and answer the three questions written on it (these were on topics such as introducing yourself, talking about the most satisfying thing about your study). Then the participants were asked to evaluate how keen they were to recruit this candidate, using a five-point scale (0 = not at all, 5 = extremely). Subsequently, participants provided ratings on a five-point scale (0 = extremely low, 5 = extremely high) regarding the perceived presence of an accent (in other words, how far the speaker’s accent was from the mainstream Mandarin accent), their level of understanding of the speaker, and the quality of their relationship with the acquainted candidates.

As mentioned above, of the four candidates (all of whom were actually associates of the experimenter), two were strangers to the participants and the other two were acquaintances. Each participant interviewed all four candidates; one stranger and one acquaintance spoke with a mainstream Mandarin accent, the others spoke with a Henan accent. A total of twelve interview questions were divided into four equal groups. The sequence of four candidates, their accents, and the sets of questions each answered were counterbalanced across participants. The answers were prepared by the experimenter and were rated as reflecting a similar quality of candidate by two recruitment experts. Each answer was about 200 words.

Finally, the participants were informed of the true purpose of the experiment and received compensation. All participants expressed their comprehension of and were unconcerned with the initial lack of transparency.

### 2.2. Results and Discussion

#### 2.2.1. Manipulation of Independent and Interference Variables

A repeated two-way analysis of variance (ANOVA) on the participants’ ratings of the perception of the presence of an accent showed that there was a significant main effect of accent (mainstream Mandarin accent, *M* = 1.42, *SD* = 0.7 vs. Henan accent, *M* = 4.37, *SD* = 0.89), *F*(2, 25) = 223.62, *p* < 0.001, $\eta_{p}^{2}$ = 0.89. But there was no main effect of familiarity (stranger, *M* = 2.88, *SD* = 1.71 vs. acquaintance, *M* = 2.9, *SD* = 1.66) or accent × familiarity interaction (*Fs* < 0.31, *ps* > 0.58). These results indicate that the manipulation of the accents was effective.

In order to test whether there was a difference in understanding what the candidates said between conditions, a repeated two-way ANOVA on the participants’ ratings of understanding what the candidates said was conducted. The results showed that there was no significant main effect of accent (mainstream Mandarin accent, *M* = 4.79, *SD* = 0.41 vs. Henan accent, *M* = 4.65, *SD* = 0.48) or familiarity (stranger, *M* = 4.77, *SD* = 0.43 vs. acquaintance, *M* = 4.67, *SD* = 0.47) or accent × familiarity interaction (*Fs* < 2.47, *ps* > 0.13). These findings indicate that the accents did not affect participants’ understanding of what the candidates said. Thus, any difference in recruitment decisions across the conditions were not due to a difference in level of understanding.

To check whether there was a difference according to the participant’s relationship with the candidate (i.e., acquainted or unacquainted) across the conditions of the Henan and standard Mandarin accents, we further conducted a repeated one-way ANOVA on the participants’ ratings of their relationship with the acquainted candidates. The results showed that there was no significant difference between the mainstream Mandarin accent condition (*M* = 3.46, *SD* = 1.42) and the Henan accent condition (*M* = 3.32, *SD* = 1.18), *F*(2, 25) = 0.53, *p* = 0.46. These results indicate that any differences in recruitment decisions across the conditions were not due to the interviewer’s relationship to the candidate.

#### 2.2.2. The Difference in the Effect of Accent between Stranger and Acquaintance Conditions

A repeated two-way ANOVA on participants’ recruitment decisions yielded a significant main effect of accent (mainstream Mandarin accent, *M* = 3.36, *SD* = 0.2 vs. Henan accent, *M* = 2.96, *SD* = 0.17), *F*(2, 25) = 4.51, *p* = 0.044, $\eta_{p}^{2}$ = 0.15, and a significant main effect of familiarity (stranger, *M* = 3.06, *SD* = 0.16 vs. acquaintance, *M* = 3.27, *SD* = 0.15), *F*(2, 25) = 8.01, *p* = 0.009, $\eta_{p}^{2}$ = 0.25. For accent, participants were more likely to recruit candidates who spoke with a mainstream Mandarin accent than those with a Henan accent; for familiarity, participants were more likely to recruit acquainted candidates than strangers. There was also a significant accent × familiarity interaction, *F*(2, 25) = 4.9, *p* = 0.036, $\eta_{p}^{2}$ = 0.16. Simple effects analyses were conducted for accent at each level of familiarity. In the interviews with candidates who were strangers, the rating of willingness to recruit was significantly different between those with a mainstream Mandarin accent (*M* = 3.35, *SD* = 1.07) and those with a Henan accent (*M* = 2.77, *SD* = 0.86), *t*(25) = 2.67, *p* = 0.01; but in the interviews with acquainted candidates, there was not a significant difference between mainstream Mandarin accent (*M* = 3.38, *SD* = 1.02) and Henan accent (*M* = 3.15, *SD* = 0.88), *t*(25) = 1.19, *p* = 0.25 (see Figure 1). These results indicate that the effect of accent on recruitment decisions differed depending on whether participants faced strangers or acquaintances.

The results suggest that accent may influence recruitment decisions; participants were more likely to recruit those who spoke with a mainstream Mandarin accent than those who spoke with a Henan accent, but this effect existed only when the candidates were strangers to the interviewer; when the candidates were acquaintances, this effect was absent.

## 3. Study 2

Study 1 involved participants who did not engage in direct communication with the speaker but instead listened to pre-recorded answers to interview questions online. This method of accent perception is not a typical or commonly encountered scenario. In Study 2, however, participants had the opportunity to interact face-to-face with the speaker, providing a more realistic environment for testing the hypothesis. This experiment was intended to investigate the hypothesis in a setting that closely resembled real-life situations.

### 3.1. Method

#### 3.1.1. Participant and Associate

Twenty-six Chinese college students (24 males; all participants spoke with a mainstream Mandarin accent) took part in the experiment. As with Study 1, the participants came from one class, and there were also four associates (two of whom were strangers and the other two acquaintances of all the participants). Neither the participants nor the associates were the same ones as in Study 1.

The associates could speak with both a mainstream Mandarin accent and a Henan accent. Since the acquainted associates were familiar with each participant, to exclude the influence of their relationships, we asked the acquainted associates to rate their relationship with each participant and counterbalanced this factor across conditions (mainstream Mandarin accent and Henan accent).

#### 3.1.2. Procedure and Materials

Participants were told that there were two experiments—one aiming to study communication, the other investment decisions—and that these experiments would be carried out successively. Both the participants and the associates were blind to the real purpose of the experiment.

Participants talked with the four associates of the experimenter on the topic of four common questions (such as: what do you think is the major environmental pollution in the world today?) in leaderless group discussions. These questions have nothing to do with trust or cooperation per se; they are simply intended to spark discussion. The associate was trained before the experiment on how to speak with participants, for example, regarding expressions and responses, to ensure that the treatment every participant received was as equal as possible. Each discussion was one-on-one and lasted about 5 min. After each discussion, participants played an investment game (for the investing experiment).

In the game section, the two players were separated into different rooms, ensuring that they could not communicate with each other. The game was a single-move public goods game. Both participants had 10 yuan (Chinese currency) in their private accounts. They could take any amount of their money out and place it into a public account. Then, the money in the public account would increase by 50% and be split equally between them. The participants would play the game four times with four partners; thus, they had four private accounts. The money in the private accounts was the real payment the participants received for taking part in the experiment, so their goal should have been to maximize the money in their private accounts. According to the rules of this game, if both participants invested a large sum of money into the public account, both would profit greatly from their investment; if both invested a little into the public account, both would profit a little; if one participant invested a little and the other invested a lot, the former would profit a lot and the latter would lose money. Which strategy the participants adopted (that is, how much money they invested) was regarded as an indicator of trust [17,18].

As with Study 1, after the participants had discussed the given topic and played the game with all four associates, they were asked questions about the perceived presence of an accent, their level of understanding of the speaker, and the quality of their relationship with the acquainted participants. Finally, the participant was told the real purpose of the experiment and was paid. All the participants expressed their comprehension and accepted the initial lack of transparency.

The two independent variables in this study were accent (mainstream Mandarin accent or Henan accent) and the relationship between associates and participants (stranger or acquaintance). Before discussion, the participant and the associate were asked to draw lots to decide which accent they should speak with. The participants always drew the mainstream Mandarin accent, whereas the associates drew either mainstream Mandarin accent or Henan accent depending on the level of the independent variables. The order effect and the effect of possible differences between the associates were counteracted in the same way as in Study 1. The dependent variable was the level of investment.

### 3.2. Results and Discussion

The procedure of the analysis for the results in this study was the same as that in Study 1.

#### 3.2.1. Manipulation of Independent and Interference Variables

A repeated two-way ANOVA was conducted on the participants’ ratings of the perception of the presence of an accent, revealing a significant main effect of accent (mainstream Mandarin accent: *M* = 1.5, *SD* = 0.73; Henan accent: *M* = 4.1, *SD* = 0.89), *F*(2, 25) = 119.05, *p* < 0.001, $\eta_{p}^{2}$ = 0.88. However, there was no significant main effect of familiarity (stranger: *M* = 2.87, *SD* = 1.6; acquaintance: *M* = 2.73, *SD* = 1.48) or accent × familiarity interaction (*Fs* < 0.93, *ps* > 0.34). These results indicate that the manipulation of the accents was effective.

To evaluate whether there was a difference in participants’ understanding of what the associates said across conditions, a repeated two-way ANOVA was conducted. The results indicated that there were no significant main effects of accent (mainstream Mandarin accent: *M* = 4.87, *SD* = 0.34; Henan accent: *M* = 4.79, *SD* = 0.41), familiarity (stranger: *M* = 4.81, *SD* = 0.4; acquaintance: *M* = 4.85, *SD* = 0.36), or accent × familiarity interaction (*Fs* < 1.64, *ps* > 0.21). These findings suggest that accent had no impact on participants’ understanding of what the associates said. Therefore, any differences in the level of investment between the conditions were not attributable to a variation in participants’ understanding.

To assess whether the degree of familiarity between participants and acquainted associates had an effect across the Henan accent and mainstream Mandarin accent conditions, a repeated one-way ANOVA was performed on the participants’ ratings of their relationship with the associates with whom they were acquainted. The results revealed no significant difference between the mainstream Mandarin accent (*M* = 3.35, *SD* = 1.26) and the Henan accent (*M* = 2.96, *SD* = 0.87), *F*(2, 25) = 1.55, *p* = 0.23. These findings suggest that any disparity in the outcomes of the discussions between the acquaintance–Henan and acquaintance–Mandarin conditions cannot be attributed to differences in the relationship between participants and acquainted associates.

#### 3.2.2. The Difference in the Effect of Accent between Stranger and Acquaintance Conditions

A repeated two-way ANOVA was conducted on level of investment, revealing a significant main effect of accent (mainstream Mandarin accent: *M* = 16.6, *SD* = 4.73; Henan accent: *M* = 14.71, *SD* = 4.79), *F*(2, 25) = 5.65, *p* = 0.025, $\eta_{p}^{2}$ = 0.18, but not of familiarity (stranger: *M* = 15.62, *SD* = 5.11; acquaintance: *M* = 15.69, *SD* = 4.58), *F*(2, 25) = 0.04, *p* = 0.85. Participants invested more in partners who spoke with a mainstream Mandarin accent compared to those with a Henan accent. There was no significant difference in investment between associates with whom participants were acquainted and those who were strangers. There was, however, a significant interaction between accent and familiarity, *F*(2, 25) = 5, *p* = 0.035, $\eta_{p}^{2}$ = 0.17. Simple effects analyses were conducted for accent at each level of familiarity. The results revealed when playing the game with strangers, there was a significant difference in investment level depending on whether the associate spoke in a mainstream Mandarin accent (*M* = 17.15, *SD* = 4.64) or a Henan accent (*M* = 14.08, *SD* = 5.18), *t*(25) = 3.56, *p* = 0.002. However, when playing the game with acquaintances, there was no significant difference between the mainstream Mandarin accent condition (*M* = 16.04, *SD* = 4.84) and the Henan accent condition (*M* = 15.35, *SD* = 4.38), *t*(25) = 0.67, *p* = 0.51 (see Figure 2).

These findings indicate that accents can influence investment decisions. Participants were more inclined to invest in individuals who spoke with a mainstream Mandarin accent compared to those who spoke with a Henan accent. However, this effect was significant only when the partners were strangers. When the partners were acquaintances, the effect of accent on investment became negligible. These results agreed with the findings in Study 1.

## 4. General Discussion

These two studies explored whether the effect of accent was different when the speaker was a stranger or an acquaintance. The results suggested that individuals were less likely to trust speakers with a Henan accent than those with a mainstream Mandarin accent. But this effect existed only when the speaker was a stranger; when the speaker was an acquaintance, this effect vanished. This pattern was true in simulated recruitment (Study 1) and investment (Study 2).

### 4.1. The Effect of the Accent on the Attitude and Behavior of the Listener

The results regarding the impact of accents on listeners’ attitudes and behaviors align with previous research findings. These studies have shown that individuals speaking with an accent are often at a disadvantage in attractiveness ratings, employment opportunities, and customer service evaluations [4,5,19,20,21]. The effect of accent on personal interactions may be attributed to the information conveyed through language pronunciation, which assists individuals in categorizing others into distinct groups [22,23], even though the listeners cannot identify the speaker’s country of origin [21]. When individuals perceive someone speaking with an accent different from their own, they tend to categorize that person as an out-group member and may exhibit a preference for members of their own accent in-group [24,25]. Accented speakers are often perceived as lower-status individuals, leading to associations with lower ability and quality [26,27].

### 4.2. The Absence of the Effect of Accent When Talking with an Acquaintance

The finding that accent had no significant effect when talking with an acquaintance is congruent with previous research.

The reduction or disappearance of the effect of accent on the attitude and behavior of the listener when speaking with an acquaintance might be due to the fact that familiarity promotes communication, even with those who speak with a Henan accent, thus reducing prejudice against them. Previous studies have found that communication between acquaintances is easier and more efficient than between strangers. Savitsky et al. found that familiarity led people to overestimate how well they communicated and led to more confidence in their judgment regarding ambiguous situations [15]. Friends can interpret each other’s thoughts and feelings more accurately than strangers [28]. Lawry et al. also found that familiarity can facilitate communication through intuition [29].

An alternative reason is that communication between acquaintances relies less on accents and more on other information, compared to communication between strangers. An accent serves as a recognizable way of pronouncing words in a language, providing information about one’s nationality, place of origin, and social role [30]. But when it comes to acquaintances who already more-or-less know this information, the effect of accent is diminished. As seen in the present study, when communicating with acquaintances rather than strangers, participants may base their recruitment and investment decisions more on their knowledge of their interlocutor than on their accent.

### 4.3. Limitations and Future Directions

There are some unexplored details and imperfections in our research, and some related questions need further research. First, the two studies tested our hypothesis only in simulated recruitment and investment situations; exploration of different scenarios would contribute to this field. Second, the participants’ odd feelings upon hearing their classmates who were also associates speak with a Henan accent was not completely eliminated, which reduces the credibility of the study’s conclusions. Future research should explore this effect in circumstances where speakers naturally exhibit a regional accent. Third, the effect of accent and its disappearance between acquaintances could be studied from the perspectives of the different stages of life, such as childhood, adulthood, and old age. Fourth, all the participants were Chinese college students. Given that there might be cultural differences in prejudice [31,32] and social interactions [33], the impact of accent when talking with strangers and acquaintances might be different in other cultures.

## Figures and Tables

**Figure 1 behavsci-14-00430-f001:**
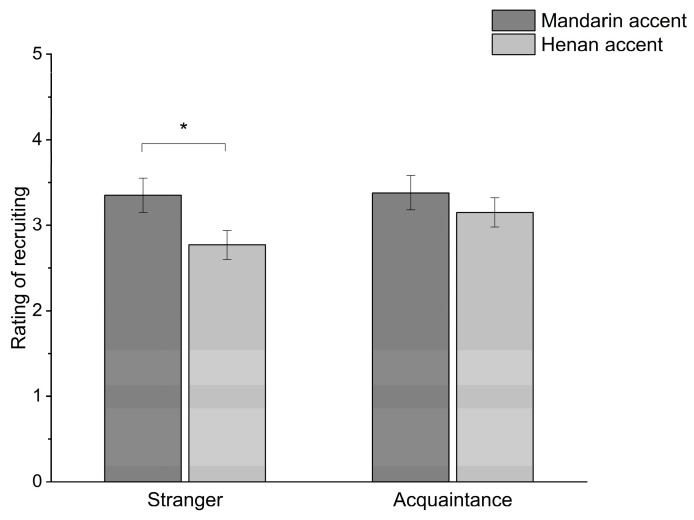
The interactive effect of accent and familiarity on recruitment decisions in Study 1. * *p* < 0.05.

**Figure 2 behavsci-14-00430-f002:**
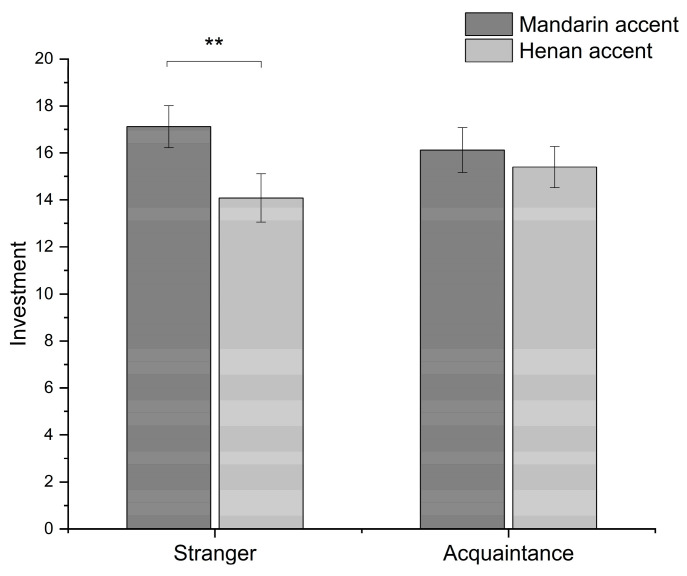
The interactive effect of accent and familiarity on investment in Study 2. ** *p* < 0.01.

## Data Availability

The data and materials are available in https://github.com/Exp123abc/Exp (accessed on 5 May 2024).

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
