# Peer review of "Familiarity Determines Whether Accent Affects Attitudes and Behaviors of the Listener"

_behavsci, 2024, doi:10.3390/bs14060430_

Round 1

Reviewer 1 Report

Comments and Suggestions for Authors

Dear Authors

The paper studies how accent has an impact on the behavior and attitudes of the people who hear it. First of all, I would like to congratulate you for the interesting subject matter, the well-structured text and the methodological development of the research. In this last sense, I agree with the authors in most of the limitations they ascribe to their work.

I consider that the article can be published in the journal because it has academic resonance, however, beforehand, we recommend that the following suggestions be taken into account:

1. Since the authors indicate several times throughout the paper that they want to analyze how the accent has an impact on social interaction it would be important that they define what they mean by this term, without leaving this important aspect of their work open.

2. It would also be interesting if the authors were more exhaustive in the description of Mandarin and Henan accents (key elements in the work), explaining in more depth their features and the socio-cultural contexts in which they are used and articulated. In this sense it is also interesting that the authors stress (although they already point out) that the results of the work are limited to the social interaction between these two specific types of accents.

Reviewer 2 Report

Comments and Suggestions for Authors

I read this paper with interest.    Some minor editing is needed to help the narrative flow, but I think that the article makes useful contribution to debate. 

Page 2 - The third paragraph merits some editing:   'Previous studies have suggested....Freeberg, for instance, has indicated'.

Page 2 - 'The participants came from one class' - College class I take it, and not social class.

Is sufficient attention given to potential differences between accent and dialect?

An interesting read.

Reviewer 3 Report

Comments and Suggestions for Authors

This manuscript is well-written and has a great focus. I laud the fact that this is an experimental method, probably the best way to address language attitudes. I like the overall review of literature, though have just a suggestion there, and I like the main idea of the study.

 However, I have several questions (or even concerns) about the operationalization of the study. It is unclear from the review of literature why we should be concerned about Henan accents as opposed to the mainstream “Mandarin” accent (though, if the Henan accent is just an *accent*, is it not also “Mandarin,” and does comparing it to “Mandarin” treat it as inferior?). Is there, for example, some stigma related to the Henan accent. In the methods, it is unclear if the “familiar” individuals are those who have been in the classroom with others the entire time speaking mainstream Mandarin accent? If to, wouldn’t the classmates know that they are just pretending to speak Henan accent? And do they even speak the accent well? Finally, I feel the experimental manipulation for Study 2 makes sense; for Study 1, I am also wondering if the entire response measure after the interviews only has 3 or 4 questions—one for each item measured. If so, this is not a very rigorous measure, and there is no way to determine reliability for any of the measures.

 With that in mind, here are my line-by line comments:

 GRAMMAR/WRITING: There are very few comments here, as the MS is well-written. There are no line numbers for me to refer to, so I will just refer to p. numbers.

·         1: “the effect of accent…was not present” change verb to singular

·         4:  “When interviewing stranger candidates…” –dangling modifier

·         5: “was reduced to insignificant” –unusual wording—assumes this statistic was first significant and then became insignificant.

·         5: “The confederate speaks”—probably past tense, like the rest of the verbs describing the study

·         5: “In the game… to rule out the influence of interactions”: unusual wording.

·         5: “these experiments would be carried out at a time”: I’m not sure what you mean, here.

·         6: “depending on the level of the independent variables due to the purposeful arrangement”: Unusual wording.

·         8: “reduced to insignificance”—similar question as above. Perhaps, more directly, “was not significant”?

 SUBSTANTIVE comments:

·         1-2: There is good coverage of several studies, though often simply placed side by side (here’s what one study says; here’s what another study says). Maybe consider from previous work some of the various variables that studies have found to *influence* accent attitudes. Then you can narrow to familiarity. Otherwise, the shift to that focus seems a bit sudden.

·         2: “people were less likely to have pleasant conversations . . . because they avoided pleasant conversations as a self-fulfilling prophecy”—the meaning here was not clear to me.

·         2: “familiarity presence may be associated with less flexible social groups”: I’m not sure what you mean by “less flexible social groups.

·         2: The focus is on the Henan accent. But why? Is it an accent near to the region of most of the participants? Does the accent have some stigma attached to it? For example, much of the accent literature seems to access prejudice of outgroups as illustrated through the accent. Are Henan-speakers an “outgroup” simply because they do not speak mainstream accent Mandarin? Maybe specify briefly the way(s) in which Henan accent is non-standard. Is it only accent, or is it a “dialect” with different grammar and word choice? If Henan is a dialect, then it would make more sense to compare it to “Mandarin.” If it is just an accent, it does not, as it is still Mandarin, just with a (different) accent (from standard Mandarin).

·         2: “non-accented speech”: This is one of the phrases (aside from comparing Henan *accent* to Mandarin *accent* that I think is problematic. It centers a particular type of speech as having “no” accent. But *all* speakers of any language have an accent. What these usages do is serve to privilege one way of speaking as “superior” to others. Of course, there might be a government mandate that requires that you do so, but I still find the word choices problematic. “Standard Mandarin,” your usual usage, works well for your purposes, I think.

STUDY 1

·         3: Method: “Thus, according to the suggestion, the available resources, and the possible exclusion, the sample sizes of the two present studies was 26.” But I thought the first line said that there were 36 participants. Are the four confederates from the class in addition to the 26 (or 36)?

·         3: Am I correct in assuming that the entire measure of four variables in the study consists of four single items?  This is “okay,” but one cannot determine reliability or validity of the measures, as far as I know. Some measure, even if brief, of each construct that would allow reliability testing would be stronger.

·         3: For the participants who spoke with the Henan accent, I had some questions. Are Henan speakers common at this university? Especially for those who are known to the students (in the same class), are they just pretending to speak a Henan accent? Do the other students realize that they are just pretending to speak the Henan accent? (If so, wouldn’t that impact the results?)

·         3: Each of the 12 questions was answered by each candidate twice? In the same interview? Provide a bit more clarity here.

·         3: RESULTS: There is a “significant main effect” of ratings of accent. What is this main effect of (what’s the question)? Is an overall rating? The “extent to which they would recruit” the candidate? In either case, it is surprising that standard Mandarin speakers were rated so much lower than Henan speakers (means of 1.42 versus 4.37). This seems to contradict other findings—so I recommend you name the point of comparison here and provide a plain English summary of the finings, as you do the other variables.

·         4: “To check whether there as a difference in the participant’s relationship with the acquainted candidate between the accent and Mandarin conditions”: Aside from assuming that “Mandarin” has no accent and that Henan speakers are not speaking Mandarin, the wording at the end of the phrase is unclear.

·         4: Figure: I see the figure is making the same distinction between “Mandarin” versus “accent” that I note above. If you change this to be either more precise or more inclusive, do so throughout the study.

·         5: 36 students from one class. Is this the same class as the first study? If so, might they be alert to the study? Also, are these students from one of the authors’ own classes? If so, do they really have “free will” to participate in the study?

·         5: “The confederate speaks in both standard Mandarin and a Henan accent.” Clarify—in the same room? Or do you mean that these are accents the confederate speaks naturally (e.g., is the confederate from Henan?).  My question about known confederates suddenly speaking Henan if this is not the dialect their classmates have always heard them speaking from the first study applies here.

·         Other than that, the experimental design of this study seemed clearer and solid.

·         6: My question about the first reported “main effect of accents” from first study applies here as well.

·         8: Good discussion overall, with a solid consideration of limitations and future directions.

Comments on the Quality of English Language

The English is fine, except for wording at some places. There are no issues with grammar, punctuation, or spelling.

Round 2

Reviewer 3 Report

Comments and Suggestions for Authors

In response to my comments on the first version, there is a clearer justification for the study of Henan accents in terms of their difference. The new explanation of the Henan accent (2:38-48) is clear and understandable. See note in line-by-line comments regarding this material.

There is more careful use (except in one case) to treat Henan as an accent of Mandarin, not a dialect or a separate language from Mandarin. This addresses one of my earlier concerns with the ideological centering of Mandarin as a “better” language that I perceived in the first version of this manuscript. There is still use throughout the MS of a “Mandarin” accent instead of a “mainstream Mandarin accent.” The text suggests that the Henan area speaks Mandarin, but with a Henan accent. So it seems that Mandarin is the language, but with different accents spoken. I recommend the authors consider, throughout the essay, referring to the Henan versus the *mainstream* Mandarin accent, rather than referring to a “Mandarin accent.” But perhaps this is the common usage in the Chinese language literature, so I will leave this choice up to the editors.

The review of lit is solid and appropriate. If the authors do revise, one other source I am aware of that looks at different perceptions of accents as they relate to hirability of a potential employee is the following:

Goatley-Soan, S., & Baldwin, J. R. (2018): Words apart: A study of attitudes towards varieties of South African English accents in a United States employment scenario. Journal of Language & Social Psychology, 37, 692-705. doi: 10.1177/0261927X18800129

Finally, the methodology seems sound, though I do have one lingering question, which appears last in the comments and might be addressed through discussion of limitations and future research.

The writing is very clear, with only a couple of things for me to comment on. There are also not a lot of substantive comments. Here are my comments page-by-page.

2:9, 27: “What enhances…” “These studies suggest that…” Consider present tense for general claims, using past tense only for specific study findings (for clarity).

2:15: “Individuals who display a robust sense of ethnic identity…” It would be helpful here to know which country these participants are from, as the results of the study may only pertain to that country.

2:17: “Besides these…” Perhaps move this transition to begin the next paragraph.

2:41: “The same vocabulary, grammar, and syntax as Mandarin.” Of course, because the Henan accent *is* Mandarin, just with a different accent, as you say elsewhere in the rev of lit. Maybe “as the mainstream Mandarin accent.”

4:38/7:27 : There are a couple of “dangling participles” (specifically, when there is a “gerund” clause: “When interviewing…” that expects the one doing the action to be the subject of the sentence, but in fact with a different subject. To fix these, say instead, either say, “When interviewing. . . participates rated” OR “In the interviews with stranger candidates…”

6:13-14: “the two participants went to different rooms to play the game to prevent them from communicating”: Meaning here is not clear.

SUBSTANTIVE COMMENTS

Rev of Literature overall is clear and well-argued.

·         2:29-31: Savitsky et al. Just out of curiosity, did Savitsky et al. have any findings that corroborated their claim?

·         2:38-49: Excellent summary of differences between Henan and Mainstream Mandarin accents. The new text also addresses my previous question as to whether there is any established stigma toward the accent. But I wondered if the fact claim, “There was no obvious stigma related to the Henan accent” has any academic evidence to support it (none is cited). If there is none to date, perhaps “Research has reported no obvious stigma related to the Henan accent.”

Method/Findings

·         Participant choice and main method (interview) is clear, as are the choice of four confederates (two familiar and two strangers.

·         3:49-4:3: This section was not clear to me. Specifically, did each candidate answer the questions twice with *each* participant, so that each participant heard the same candidate speaking with both a Mainstream Mandarin accent and a Henan accent.

·         4:6/6:45 :Consider saying “ratings of the perception of a presence of accent,” just to remind the reader that the “ratings” here regard if an accent was present. Other findings here make sense and have good, plain-English statements of what each finding means. The summary paragraphs at the end of coverage for each study are clear and helpful.

·         5:16: Are these the same 26 students as the first study? This is important to for the reader to know, as it might introduce a response bias if the participants have any memory of the first experiment.

·         6:9: Briefly explain the relationship between the four questions and the game. For example, are these questions ostensibly to allow the participants to decide if they “trust” the other person in the game? Are the confederates answers to the questions pre-arranged to add control to the input of the question answers on the playing of the game?

·         7:1ff: A strength of the study is the careful ruling out of alternative influencing factors in the results sections of each study and, again, clear summary statements of fidnings.

·         7:19ff: The results seem to contradict: The first statement is that there is “a significant main effect” between “level of familiarity.” The means are 15.62 and 15.69—very close, with a follow-up statement that “there was no significant difference between acquainted and strange partners” (7:23-24).

Discussion:

·         8:23: “tend to categorize that person as an out-group member…” I think that the study findings work well to show that this effect is true only if the stranger is speaking with the Henan accent (assuming that Henan speakers are frequent or regularly present at the university where the study is done, something I don’t recall reading from the background of the study). One question I have that remains and that the limitations should consider is that, in this study, classmates with whom students are already familiar and who they likely know do not normally speak with a Henan accent are suddenly speaking with a Henan accent in these experiments. Wouldn’t the students find that strange? In my country, it might be like having a classmate suddenly start speaking with a “Southern” or “Boston” accent in an experiment. It might be that the results are different with strangers because the participants assume that they really speak with a Henan accent, but not with their classmates, because they know that they are just acting.

If there is not a large number of Henan-accent speakers at the university, the participants might also assume that the unknown confederates are acting. However, the findings do not address whether having an acquaintance who *regularly* speak with a Henan accept would make a difference in hiring or investment decisions. Future research could include classmates who naturally speak with a Henan accent and those who don’t in the “acquaintance” condition.
